# Estimation of Shoulder Joint Rotation Angle Using Tablet Device and Pose Estimation Artificial Intelligence Model

**DOI:** 10.3390/s24092912

**Published:** 2024-05-02

**Authors:** Shunsaku Takigami, Atsuyuki Inui, Yutaka Mifune, Hanako Nishimoto, Kohei Yamaura, Tatsuo Kato, Takahiro Furukawa, Shuya Tanaka, Masaya Kusunose, Yutaka Ehara, Ryosuke Kuroda

**Affiliations:** Department of Orthopaedic Surgery, Kobe University Graduate School of Medicine, Kobe 650-0017, Japan; shunsaku@med.kobe-u.ac.jp (S.T.); m-ship@kf7.so-net.ne.jp (Y.M.); hanakoni@med.kobe-u.ac.jp (H.N.); koheidesuyo@yahoo.co.jp (K.Y.); t.kato.ort@gmail.com (T.K.); takahiro.0412.0321@gmail.com (T.F.); shuyatanaka0517@gmail.com (S.T.); mkunose@med.kobe-u.ac.jp (M.K.); yutakaehara0720@gmail.com (Y.E.); kurodar@med.kobe-u.ac.jp (R.K.)

**Keywords:** artificial intelligence, shoulder, range of motion

## Abstract

Traditionally, angle measurements have been performed using a goniometer, but the complex motion of shoulder movement has made these measurements intricate. The angle of rotation of the shoulder is particularly difficult to measure from an upright position because of the complicated base and moving axes. In this study, we attempted to estimate the shoulder joint internal/external rotation angle using the combination of pose estimation artificial intelligence (AI) and a machine learning model. Videos of the right shoulder of 10 healthy volunteers (10 males, mean age 37.7 years, mean height 168.3 cm, mean weight 72.7 kg, mean BMI 25.6) were recorded and processed into 10,608 images. Parameters were created using the coordinates measured from the posture estimation AI, and these were used to train the machine learning model. The measured values from the smartphone’s angle device were used as the true values to create a machine learning model. When measuring the parameters at each angle, we compared the performance of the machine learning model using both linear regression and Light GBM. When the pose estimation AI was trained using linear regression, a correlation coefficient of 0.971 was achieved, with a mean absolute error (MAE) of 5.778. When trained with Light GBM, the correlation coefficient was 0.999 and the MAE was 0.945. This method enables the estimation of internal and external rotation angles from a direct-facing position. This approach is considered to be valuable for analyzing motor movements during sports and rehabilitation.

## 1. Introduction

The shoulder is composed of three bones (clavicle, scapula, and humeral head) and four joints (glenohumeral, sternoclavicular, acromioclavicular, and scapulothoracic joints). The glenohumeral joint is the primary shoulder joint existing between the head of the humerus and the glenoid fossa of the shoulder girdle. The glenohumeral joint is a ball-and-socket joint, with only 25% of the humeral head fitting into the glenoid fossa, forming a very shallow joint. The shoulder is the most flexible joint in the body and is also an unstable joint [1,2]. The normal range of motion of the shoulder is 150 to 180 degrees of forward flexion, 40 to 60 degrees of extension, 150 to 180 degrees of abduction, 60 to 90 degrees of external rotation, and 50 to 70 degrees of internal rotation [3,4,5]. For each movement, a basic axis and a movement axis are defined, so it is necessary to know these when measuring angles.

Motion analysis of the shoulder joint is an important element in the field of sports medicine and rehabilitation [6]. The gold standard for shoulder joint angle measurement is the use of a universal goniometer, but it is dependent on the evaluator’s knowledge and skill [7]. Other measurement methods have been reported using visual inspection, inclinometers, smartphone applications, or markers. However, the reliability of visual inspection (ICC: 0.15~0.92), inclinometers (ICC: 0.5~0.95), and smartphone applications (ICC: 0.38~0.99) varies from report to report, and analysis during exercise is difficult [8,9,10,11,12]. Motion capture systems using markers and high-speed cameras are constrained by cost and environment [13]. Therefore, it is useful to develop a method of angle measurement that is both accurate and inexpensive.

With the spread in the use of smartphones, many physicians have smartphones. Furthermore, the advancement of smartphone cameras has made it possible to record high-quality videos. Therefore, the use of smartphones for medical photography is becoming increasingly popular and appears to be accepted by many patients [14]. If angles can be estimated from videos taken with a smartphone, there is no need for special equipment and no cost.

Recent advances in AI have led to the development of various posture estimation AI models. For example, a driver monitoring system (DMS) can detect a driver’s driving posture and facial expressions to reduce the risk of accidents [15,16]. Pose estimation AI models also include MediaPipe, Yolov8, and PoseNet. The Yolov8 is the latest model in the Yolo series that serves as a general model for visual understanding. The Yolov8 is the most recent model in the Yolo series. It performs better than previous versions in terms of accuracy and speed [17]. The PoseNet is a real-time pose estimation model developed by Google that detects 17 body parts and uses deep learning techniques to estimate human posture in real time from both photos and videos [18]. MediaPipe, developed by Google, is an open-source, cross-platform system that specializes in real-time media processing, especially video and image analysis and processing. MediaPipe is capable of tracking the location of 33 landmarks on the human body and measuring the position coordinates of the landmarks [18]. By focusing on detecting bounding boxes for relatively rigid body parts, this method uses a minimally sufficient number of landmarks for the face, hands, and feet to estimate the rotation, size, and position of the region of interest for subsequent models. Compared to other posture estimation methods, MediaPipe has demonstrated superior accuracy, but it is not perfect [19]. It has been reported that the integration of MediaPipe with machine learning models for posture analysis has improved accuracy in assessing shoulder abduction angles [20].

In this study, we attempted to develop a highly accurate method of angle estimation that does not require special equipment and is inexpensive. As a method for angle estimation, we assumed that the angle could be estimated with high accuracy by combining MediaPipe with machine learning. In addition, the rotational movement of the shoulder joint was selected as the movement for angle estimation. The basic axis of shoulder rotation is a vertical line to the anterior plane through the elbow, and the movement axis is the ulna. Therefore, the shoulder rotation angle is difficult to measure from a face-to-face position. In addition, although there have been reports of shoulder angles measured using posture estimation AI, there have been no reports of rotation angles measured [21,22]. We examined the possibility of estimating the rotation angle of the shoulder joint by detecting the coordinates from the video using MediaPipe and combining it with machine learning.

## 2. Materials and Methods

### 2.1. Participants

To evaluate the range of motion of the shoulder rotation angle, 10 healthy adult volunteer subjects were involved (10 males, mean age 37.7 years, mean height 168.3 cm, mean weight 72.7 kg, mean BMI 25.6). They were instructed to perform shoulder joint external and internal rotation exercises in a standing position facing forward, with the right upper extremity drooped and the elbow joint in a 90-degree flexion. This study was approved by the Kobe University Ethics Committee (approval number: B210009), and informed consent was obtained from all participants.

### 2.2. Angle Measurement Application

In this study, a smartphone angle measurement application was used as the true angle, and the accuracy of angle estimation was examined. The angle measurement application used was Measure Angles-Bubble Level (ver. 3.99.90, JRSoftWorx), and the reliability of the application was examined. To evaluate the reliability of the application, lines of 50°, 30°, 0°, −30°, and −50° angles were marked using a protractor on the tabletop. The 30° internal rotation is indicated as −30°. We validated the accuracy of the app when the smartphone was placed on a tabletop and when it was fixed to the forearm. The smartphone was fixed to the subject’s forearm with a band. With the application active, the forearm was moved over the line for each angle and the angle displayed on the application was recorded. The angle displayed on the application at each angle was recorded. The protocol was repeated 30 times for each mark, and the mean absolute error (MAE) was examined (Figure 1).

### 2.3. Data Acquisition and Image Processing by MediaPipe

The smartphone was attached to the volunteer’s forearm, and a tablet device (iPhone 14 Pro, Apple, Cupertino, CA, USA) was placed at a height of 150 cm from the floor, 2 m away from the subject (Figure 2). All video recording was performed by a designated examiner (S.T.). The video recording mode was 1080p HD, 30 fps, with each angle recorded for approximately 2 s. The captured video files were processed with MediaPipe pose’s phyton library to obtain each landmark coordinate (x, y, z). The landmark coordinates were normalized between 0.0 and 1.0 by the image width (x) and height (y). However, since the distance between the object and the camera is not used as a parameter in this study, the z coordinate is not used. In this study, we used the coordinates of the bilateral shoulder joints, right elbow joint, right wrist joint, and right hip joint, which are among the 33 landmarks detectable by MediaPipe (Figure 3). Vector calculations were performed using each coordinate, and angle and distance parameters were calculated.

### 2.4. Machine Learning (ML)

In this study, we compared models created by machine learning with five different methods (linear regression, ElasticNet, SVM, random forest regression, and Light GBM). Traditional linear regression and ElasticNet were adopted as the basic regression methods. SVM is an algorithm that performs classification or regression by determining a boundary or hyperplane that separates two classes of data. Random forest regression is an ensemble learning process that uses multiple decision trees to obtain high reservation performance. Light GBM is based on the decision tree algorithm and requires less learning time and memory usage than traditional methods [9]. Figure 4 shows the workflow of this study. Ten volunteers were filmed at every 10 degrees of rotational angle from -50 to 50 degrees, and 11 types of data were obtained. The internal rotation direction was defined as negative values and the external rotation angle was defined as positive values. A total of 10,608 images were obtained from the video data. The videos taken were randomly divided into training data for machine learning (80%) and test data used for angle estimation (20%). After determining the optimal parameters for each ML algorithm from the training data, the correlation coefficient and MAE were determined as performance indicators to compare the accuracy of the models.

Another method of assessing the quality of a regression analysis is the residual plot. The residual plot displays the difference (residuals) between the predicted and actual values in the regression analysis. For each ML model, the actual value (true angle) is plotted on the *x*-axis and the residual (actual angle–predicted angle) on the *y*-axis. A residual close to zero indicates that the model adequately captures the data. In this study, residual plots for linear regression, ElasticNet, SVM, random forest regression, and Light GBM were created to evaluate the accuracy of each model.

Feature importance and SHAP (Shapley additive explanations) values were used to visualize important parameters for estimating shoulder rotation angle. The feature values were normalized by the total of all features present in the tree; the overall importance of a feature was obtained by dividing it by the total number of trees in the ML. In addition, the contribution of each feature to the prediction was evaluated using the SHAP value. Based on game theory, the SHAP value was defined as the contribution of each feature to the model’s predictions. SHAP values are useful for increasing the interpretability of a model and are especially beneficial in complex models [23]. All analyses of ML models were performed using Scikit -learn v1.0.2 library.

### 2.5. Parameters

Vectors were defined using the coordinates of the bilateral shoulder joints, the right elbow joint, the right wrist joint, and the right hip joint, as recognized by MediaPipe, and several parameters for use in machine learning were established (Figure 5).

The parameters used in the analysis are described in detail below (Table 1).

norm_elbow_size: The value obtained by dividing the cross product of the vector from the right shoulder to the right elbow and the vector from the right elbow to the right wrist joint by the square of the length of the vector from the right shoulder joint to the right hip joint.norm_shoulder_size: The value obtained by dividing the cross product of the vector from the right elbow to the right wrist and the vector from the right shoulder to the right hip joint by the square of the length of the vector from the right shoulder joint to the right hip joint.norm_forearm_distance: The value obtained by dividing the cross product of the vector from the left shoulder to the right shoulder and the vector from the right shoulder to the right hip joint by the square of the length of the vector from the right shoulder joint to the right hip joint.norm_uparm_distance: The value obtained by dividing the distance of the vector from the right shoulder to the right elbow by the distance of the vector from the right shoulder joint to the right hip joint.elbow_angle: The angle formed by the right shoulder, the right elbow, and the right wrist.shoulder_angle: The angle formed by the right elbow, the right shoulder, and the right wrist.trunk_angle: The angle formed by the left shoulder, the right shoulder, and the right hip.

## 3. Results

### 3.1. Validation for Application

To evaluate the reliability of the angle measurement application, MAE was calculated using the angle data obtained. The MAE was 0.96 when the smartphone was placed on the tabletop and 0.91 when it was fixed to the forearm. Therefore, we used the app as the source of the true value.

### 3.2. Estimation of Shoulder Rotation Angle

Machine learning was performed using training data randomly selected from a total of 10,608 acquired images. An example of the angle estimation scene is shown in Figure 6. The estimated angle is shown above the volunteer’s head. Comparing the models, both the correlation coefficient and MAE were superior in Light GBM (Table 2). The hyperparameters used in model training are summarized in Table 3 For each ML model, the actual values (true angle) were plotted on the *x*-axis, and the residuals (actual angles–predicted angles) on the *y*-axis (Figure 7, Figure 8, Figure 9, Figure 10 and Figure 11). The mean and standard deviation of each residual of the linear regression and Light GBM models for each angle are summarized (Table 4 and Table 5). A heat map showing the correlation of each parameter is shown in Figure 12. According to the heatmap, norm_elbow_size and norm_shoulder_size showed a strong correlation with the angle. Furthermore, norm_elbow_size scored higher when evaluated using feature importance in angle estimation, indicating that norm_elbow_size has a greater impact on models that estimate shoulder rotation angle. Heat map results are consistent with feature importance and SHAP value results. From the evaluation of SHAP values, it was found that norm_elbow_size had high scores in both positive and negative aspects (Figure 13).

## 4. Discussion

Shoulder pain impairs patients’ quality of life and ability to work [24]. Therefore, shoulder motion analysis is important in medical practice [6]. In this study, the combination of machine learning and MediaPipe made it possible to estimate the rotation angle of the shoulder from a straight-forward-facing position. Normally, a goniometer is used to measure the angle of the shoulder joint. However, it has some limitations, such as the difficulty of measuring during exercise. In addition, it is difficult to measure the shoulder rotation angle from a facing posture. In previous studies, there have been some reports of angle estimation using posture estimation AI, but there have been no reports of shoulder joint rotation angles [19,20,21]. The method in this study is beneficial for angle measurement in clinical practice because it allows real-time evaluation and is not costly.

The methods of machine learning used were linear regression and Light GBM, both of which had correlation coefficients of 0.972 and 0.997, respectively, indicating high accuracy. The MAE for linear regression was 6.056, indicating an error of more than 5 degrees, but it is expected that this can be improved by increasing the number of cases in the future. Linear regression defines a linear relationship between variables x and y with the equation y = a + bx and allows the value of the dependent variable y to be estimated from the independent variable x [25]. In this study, the linear regression model was able to estimate the angle of rotation with high accuracy. It is thought that this was due to the parameter settings (especially norm_elbow_size) having a linear relationship with the angle of rotation. There have been previous reports examining posture estimation using a combination of machine learning with MediapPipe and Light GBM, but none have mentioned the shoulder rotation angle [26,27]. In this study as well, the estimation of angles was highly accurate with a correlation coefficient of 0.997 and MAE of 1.464. It is a cost-effective method for estimating angles, as it requires minimal computation and no special equipment.

Residual plots are essential for evaluating regression models by clarifying issues related to assumptions and pinpointing outliers. From the Light GBM plots in this study, we observe a wide spread of residuals across all angles, with no systematic deviations. Notably, horizontal bands within the residuals may suggest consistent prediction errors at certain angular values. Linear regression plots also show a similar lack of systematic patterns, but the spread in the Light GBM suggests a more nuanced understanding of variance, which may lead to superior accuracy. In both models, residuals are centered around zero, generally suggesting the reliability of predictions.

SHAP values explain the effect of having a certain value for a given feature in comparison to the prediction we would make if that feature took some baseline value. The features listed on the *y*-axis are the features that the model considers when making estimates. The position on the *y*-axis is determined by the importance of each feature, with the higher-ranked features having a greater impact on the angle estimation model output. The parameters such as norm_elbow_size seem to be one of the most impactful features, as it is located at the top of the *y*-axis. Features with SHAP values to the right (positive impact) increase the model’s output, while those to the left (negative impact) decrease the output. Figure 10 shows that the higher the value of norm_elbow_size (indicated by pink/red on the right side), the higher the predicted value, and the lower the value of norm_elbow_size (indicated by blue on the left side), the lower the predicted value. Norm_elbow_size is a value normalized by dividing the area of the parallelogram formed by the upper arm and the forearm by the square of the trunk length. Since the length of the trunk does not change with movement, normalization of the trunk can accommodate changes in the distance between the camera and the subject. Furthermore, the area of the parallelogram increases as the angle of gyration increases from a position facing directly forward (Figure 14). There is a negative correlation with internal rotation and a positive correlation with external rotation. This relationship occurs because the direction of internal rotation is defined as minus. Therefore, the correlation between the parallelogram and the angle of rotation would change from a position where the camera and the subject are not directly facing each other. Some reports have previously changed the camera position to estimate the shoulder abduction angle. It has been reported that if appropriate parameters are set during machine learning and the learning is carried out, it is possible to estimate the angle with high accuracy [21].

The estimation model developed in this study can display angles in real time without the use of complicated devices; this is a feature that was not seen in previous studies and it is considered innovative. With a tablet device, real-time examination and measurement can be performed remotely, expanding the potential for applications in telemedicine and rehabilitation. The possibility of analyzing the movements of thumbs and other fingers from various positions with a smartphone is expected to contribute to the efficiency of motion analysis and its application to telemedicine and rehabilitation.

There are several limitations to this study. Firstly, the maximum external rotation angle with the arm at the side is defined as 60 degrees, and the maximum internal rotation range is 80 degrees [28]. However, some subjects in the present study had difficulty actively setting the external rotation angle to 60 degrees, so the rotation angle was limited to 50 degrees. Second, the application is used as the true angle. The accuracy of the app varies according to the report [10]. In this study, the reliability assessment of the app is in 10-degree increments, and the evaluation of the rotational angle is also in 10-degree increments. In actual rehabilitation and sports sites, a more detailed angle evaluation may be necessary. Third, it only evaluates the rotational angle in the position with the arm at the side. The shoulder joint is a complex motion joint with three-dimensional flexion–extension and abduction–adduction, and it is necessary to be able to estimate the angle of more complex motions. Future research on angle estimation models created by collecting data on more complex motions is needed.

## 5. Conclusions

This study demonstrated the potential of estimating shoulder joint rotation angles by combining MediaPipe and machine learning. The machine learning model was trained on images from videos of 10 volunteers. The model was evaluated using correlation coefficients and MAE. Both linear regression and Light GBM methods were able to estimate shoulder rotation angles with high accuracy. This method allows for the real-time estimation of shoulder rotation angles without the need for special equipment, which has never been used before. Estimation is possible with only a tablet terminal and is expected to be utilized in remote areas.

In conclusion, the combination of MediaPipe and ML enabled highly accurate estimation of shoulder rotation angles in real time from videos taken with a smartphone from a forward-facing position.

## Figures and Tables

**Figure 1 sensors-24-02912-f001:**
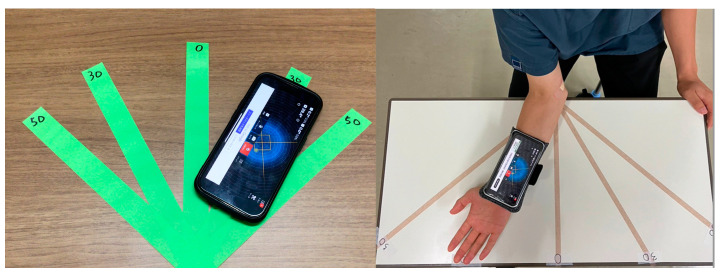
Validation of the angle measurement app.

**Figure 2 sensors-24-02912-f002:**
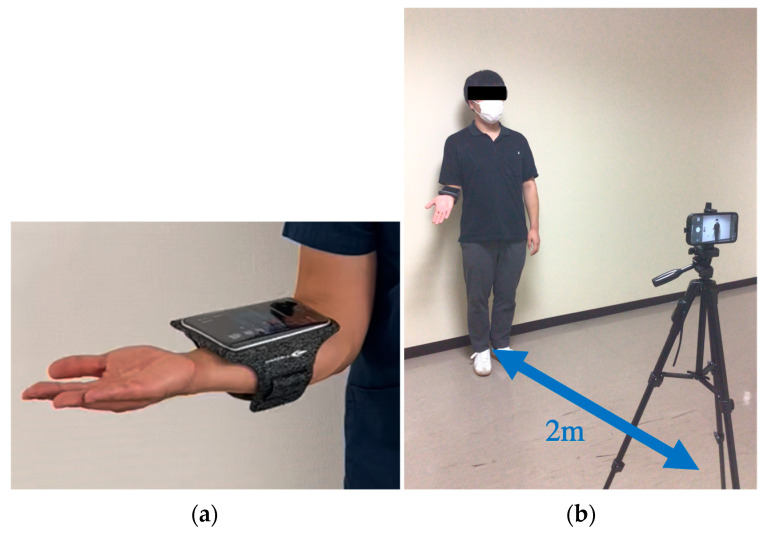
(**a**) A smartphone attached to the forearm. (**b**) The camera position for recording.

**Figure 3 sensors-24-02912-f003:**
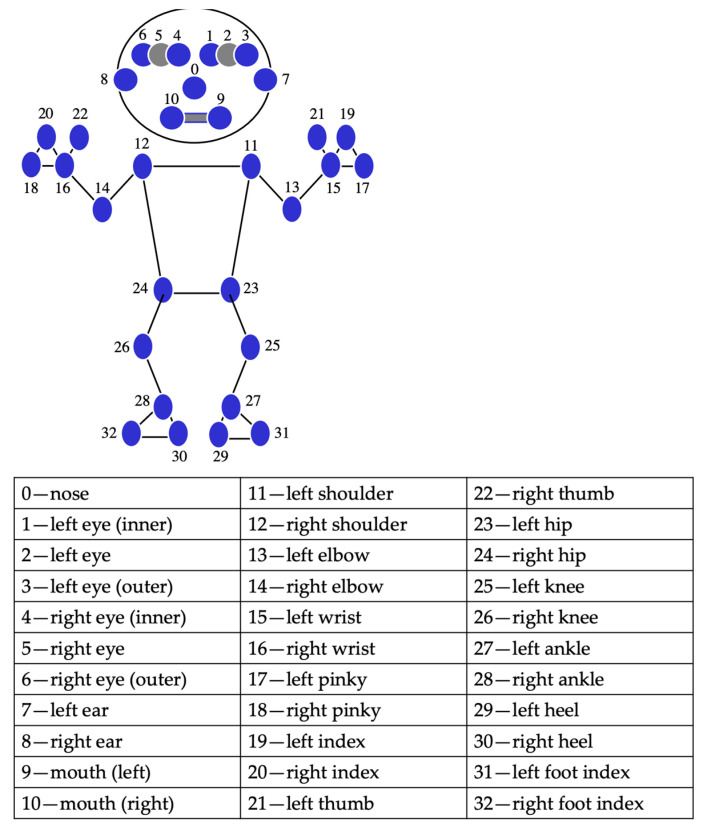
MediaPipe landmarks.

**Figure 4 sensors-24-02912-f004:**
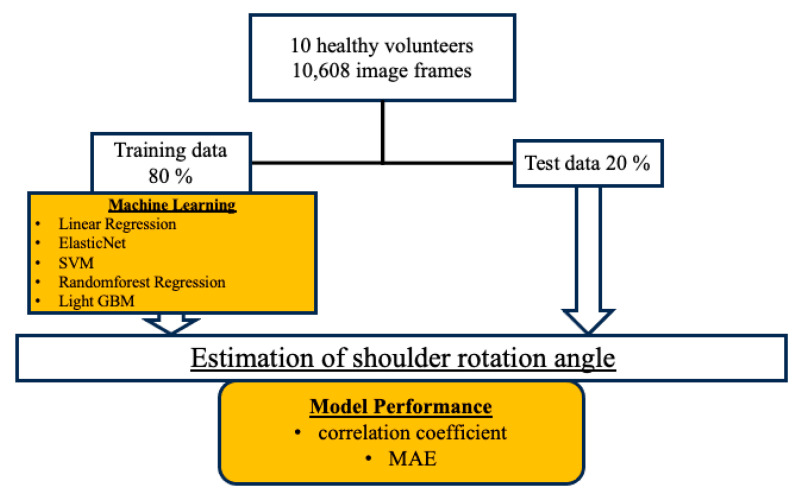
The workflow of the data acquisition and ML. A total of 10,608 images were captured at 30 fps over 2 s from 10 participants with 11 different shoulder rotation angles ranging from −50 to 50°. The captured images were randomly separated and 80% were assigned as training data to fine-tune the model parameters and 20% as test data to evaluate the efficacy of each ML model. Training data were used to identify the best hyperparameters for each ML algorithm, and MAE and correlation coefficient were used as indicators to estimate and compare model accuracy.

**Figure 5 sensors-24-02912-f005:**
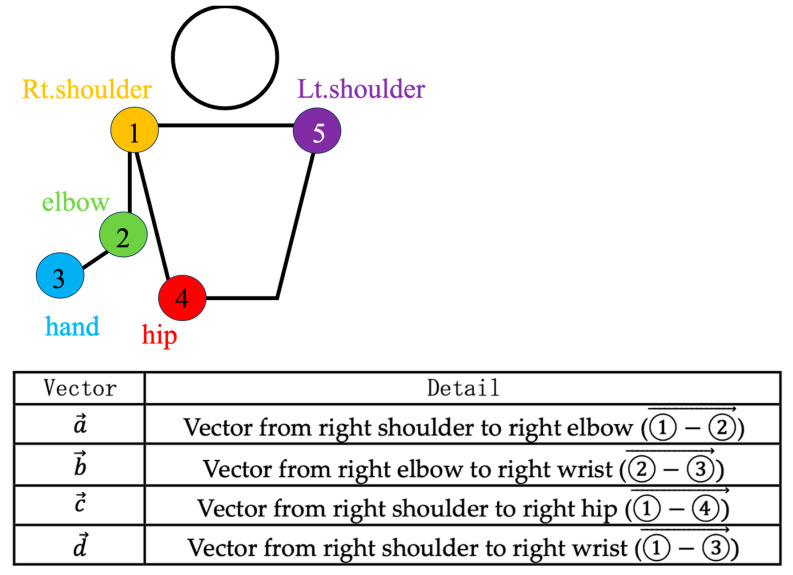
Landmarks and vectors were used in this study.

**Figure 6 sensors-24-02912-f006:**
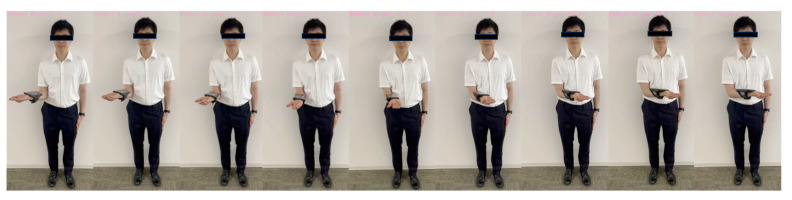
Actual angle estimation images.

**Figure 7 sensors-24-02912-f007:**
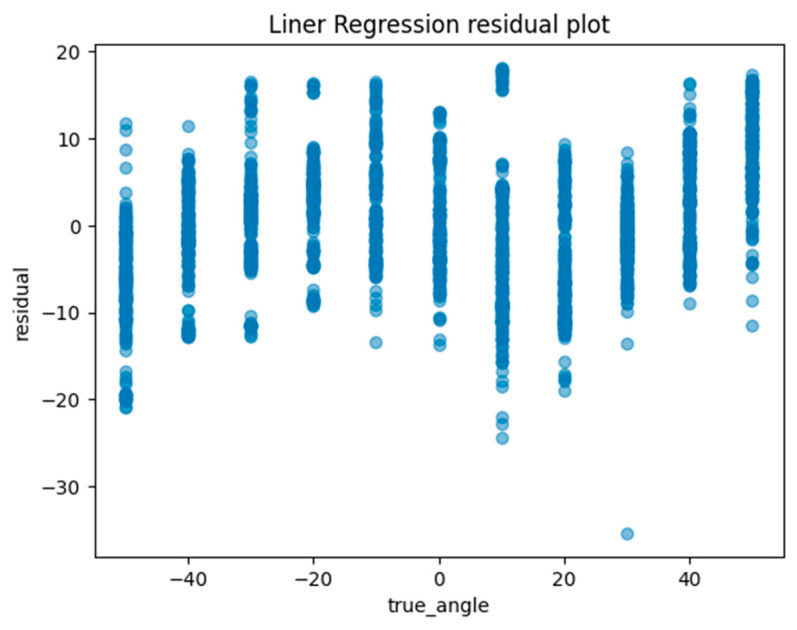
The residuals (actual angles–predicted angles) of the linear regression model were plotted and compared against the actual angles for the test data.

**Figure 8 sensors-24-02912-f008:**
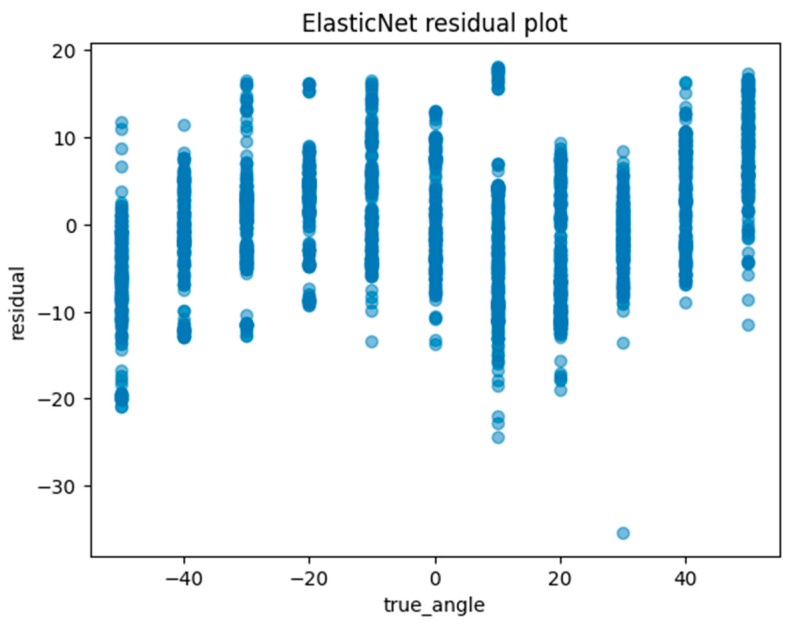
The residuals (actual angles–predicted angles) of the ElasticNet model were plotted and compared against the actual angles for the test data.

**Figure 9 sensors-24-02912-f009:**
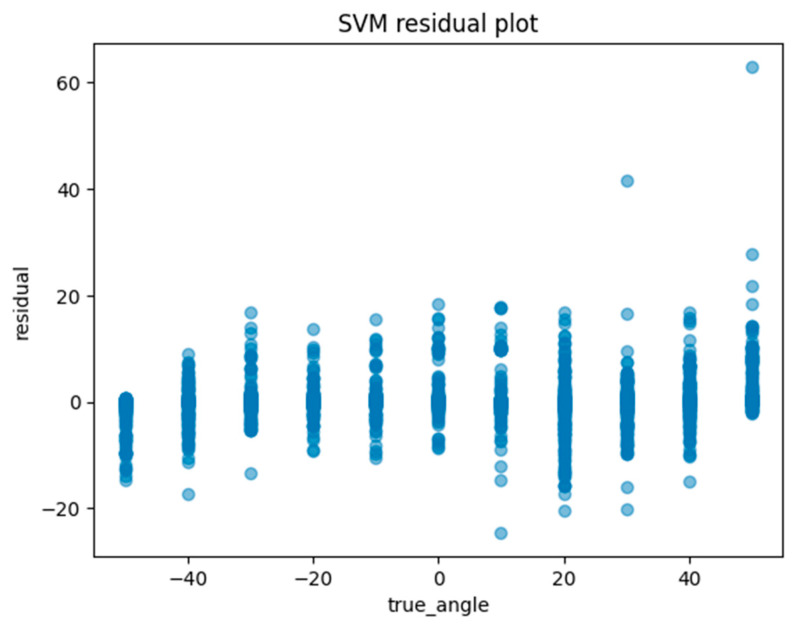
The residuals (actual angles–predicted angles) of the SVM model were plotted and compared against the actual angles for the test data.

**Figure 10 sensors-24-02912-f010:**
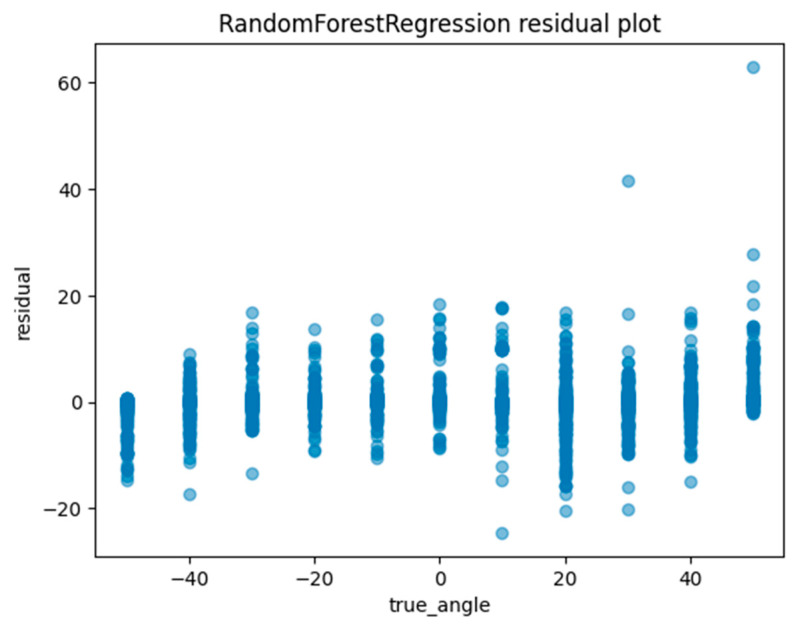
The residuals (actual angles–predicted angles) of the random forest regression model were plotted and compared against the actual angles for the test data.

**Figure 11 sensors-24-02912-f011:**
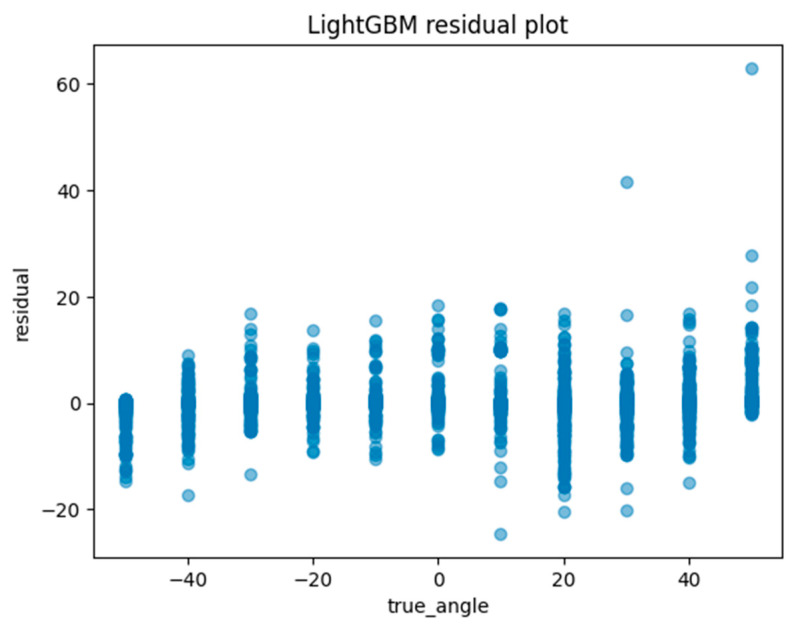
The residuals of the Light GBM model were graphically represented and juxtaposed with the actual angles for the test data.

**Figure 12 sensors-24-02912-f012:**
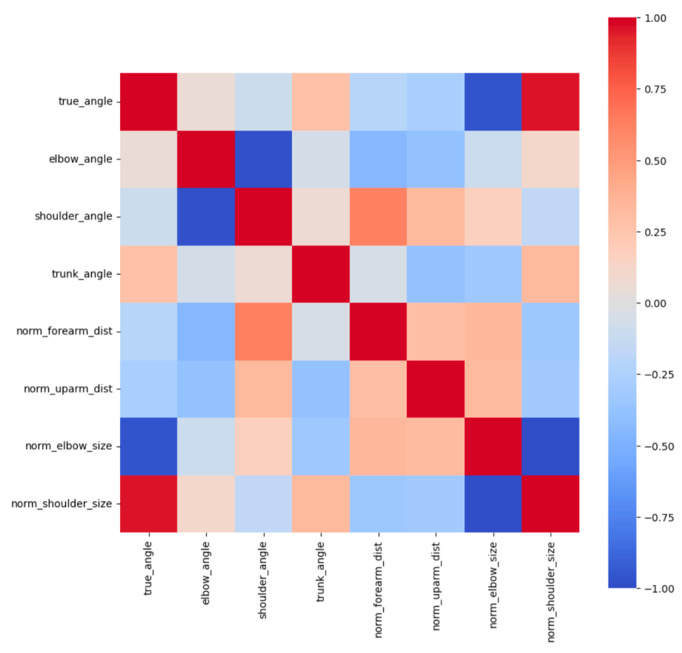
Heat map for each parameter. Warm colors show a positive correlation and cold colors show a negative correlation.

**Figure 13 sensors-24-02912-f013:**
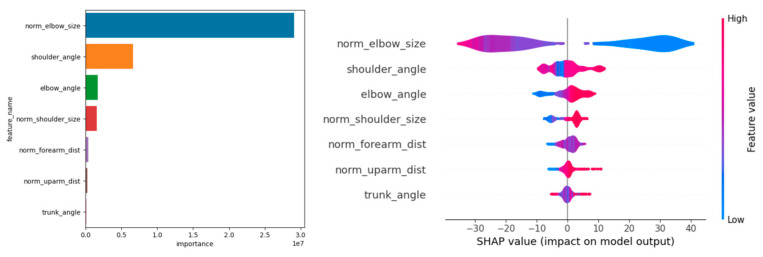
Feature importance and SHAP value in Light GBM.

**Figure 14 sensors-24-02912-f014:**
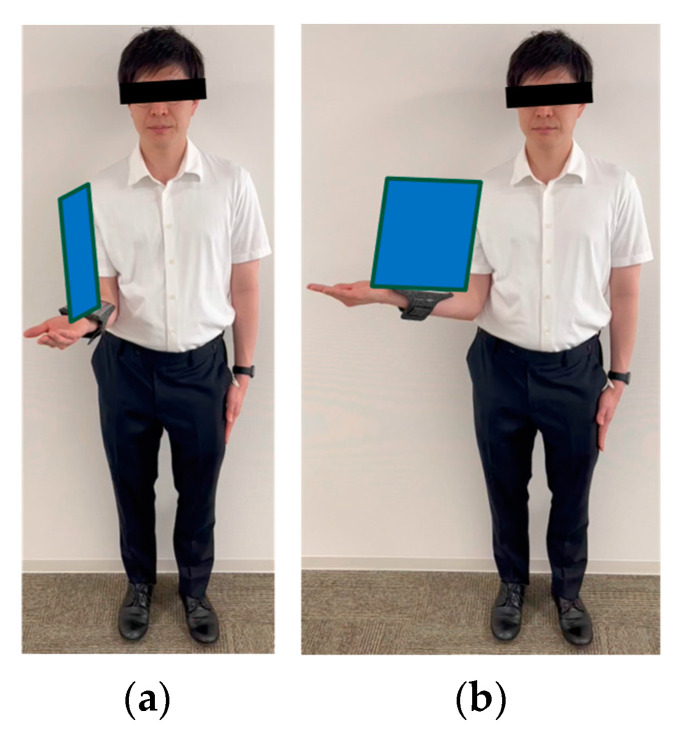
The area of parallelograms tends to increase similarly as the angle of gyration increases ((**a**) ER 10°; (**b**) ER 50°).

**Table 1 sensors-24-02912-t001:** List of parameters used in this study.

Name	Definition
norm_elbow_size	$\vec{a}\;$ $\times \vec{\;b}$ $/{\left|\vec{c}\right|}^{2}$
norm_shoulder_size	$\vec{a}\;$ $\times \vec{\;d}$ $/{\left|\vec{c}\right|}^{2}$
norm_forearm_distance	$\vec{\left|b\right|}$ $/\left|\vec{c}\right|$
norm_uparm_distance	$\vec{\left|a\right|}$ $/\left|\vec{c}\right|$
elbow_angle	∠①-②-③
shoulder_angle	∠②-①-③
trunk_angle	∠⑤-①-④

**Table 2 sensors-24-02912-t002:** The accuracies of each ML model are summarized.

	Linear Regression	ElasticNet	SVM	Random Forest Regression	Light GBM
Correlation coefficient	0.972	0.972	0.989	0.994	0.997
MAE	6.056	5.935	2.468	2.063	1.464

**Table 3 sensors-24-02912-t003:** Summary of hyperparameters of each model.

ML Model	Linear Regression	ElasticNet	SVM	Random Forest Regression	Light GBM
Representative parameters	Penalty: L2 C: 1.0 Solver: lgfbs	Alpha: 1 × 10^−5^ L1-ratio: 0.889 Fit intercept: True	C: 10.0 Gamma: 0.0046	Criterion: squared error Max depth: 6 Number estimators: 10	Objective: mean absolute error Learning rate: 0.076 Max depth: 8

**Table 4 sensors-24-02912-t004:** The mean and standard deviation of each residual for each angle of the linear regression model.

Linear Regression	Residual	Residual
	**Mean (°)**	**Standard Deviation (°)**
**True_Angle (°)**		
−50	−6.44	6.27
−40	−1.32	5.61
−30	0.39	6.10
−20	2.46	5.42
−10	2.55	6.51
0	0.63	6.25
10	−3.31	8.72
20	−4.81	6.65
30	−1.63	4.38
40	3.26	5.77
50	8.5	5.57

**Table 5 sensors-24-02912-t005:** The mean and standard deviation of each residual for each angle of the Light GBM model.

Light GBM	Residual	Residual
	**Mean (°)**	**Standard Deviation (°)**
**True_Angle (°)**		
−50	−1.88	3.47
−40	−0.59	3.47
−30	0.31	3.23
−20	0.14	2.71
−10	0.31	3.34
0	1.05	4.09
10	0.68	4.53
20	−1.19	5.96
30	−0.44	5.19
40	0.62	4.28
50	3.33	6.57

## Data Availability

The data presented in this study are available upon request from the corresponding author. The data are not publicly available because of confidentiality concerns.

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
