# Peer review of "Estimation of Shoulder Joint Rotation Angle Using Tablet Device and Pose Estimation Artificial Intelligence Model"

_sensors, 2024, doi:10.3390/s24092912_

Round 1
Reviewer 1 Report
Comments and Suggestions for Authors
This manuscript entitled “Estimation of Shoulder Joint Rotation Angle Using Tablet Device and Pose Estimation Artificial Intelligence Model” primarily aimed to estimate the shoulder joint internal/ex-ternal rotation angle using the combination of pose estimation Artificial Intelligence (AI) and machine learning model. The authors bring an interesting study, but there are still some problems that cannot up this review to a publishing level. Some suggestions are listed in the specific comments below.
Specific comments:
1. In the abstract part, line 18, “Videos of the right shoulder of 10 healthy volunteers were recorded.” Please provide detailed anthropometry information for participants, such as height, weight, and body mass index.
2. In the introduction part, the sequence of the citation should start from 1.
3. In the introduction part, line 46-47, “Other measurement methods have been reported using visual inspection, inclinometers, smartphone applications, or markers.” Please cite more references to support this statement.
4. In the introduction part, it is recommended to add the research gap and highlight the value of this study.
5. In the Materials and Methods part, line 84, “To evaluate the range of motion of the shoulder rotation angle, 10 healthy adult volunteer subjects were involved (10 males, mean age 37.7 years).” It is suggested to provide detailed anthropometry information for participants.
6. In the discussion part, it is recommended to provide a brief description of the aim and main findings in the first paragraph of the manuscript. Some recently studies could be added in the discussion, such as:
Convergent Validity of Thoracic Outlet Syndrome Index (TOSI). Physical Activity and Health, 6(1), p.16–25.DOI: https://doi.org/10.5334/paah.162
7. In the conclusion part, in the opinion of the reviewer, authors provide too much description about the method, and the reviewer suggests that the authors should focus on the main findings of this study.
Comments on the Quality of English LanguageMinor editing of English language required
Reviewer 2 Report
Comments and Suggestions for Authors
The paper attempts to estimate the shoulder joint internal/external rotation angle using the combination of pose estimation chine learning model. The authors assumed that the angle could be estimated with high accuracy by combining MediaPipe with machine learning methods.
The paper is well-written and easy to understand. However, I have the following concerns:
1. I suggest the authors to provide more descriptions of recent AI-based posture estimation methods. Some deep learning-based models are not introduced.
2. Figure 5 does not have a corresponding citation in the main text. Please check for any redundancies or omissions.
3. I suggest the authors to compare the proposed model with several machine learning methods in the experiments. Visualized comparisons should be provided in the experiments.
4. The critical parameters in the proposed model should be discussed in detail, and the corresponding experiments should be provided.
Comments on the Quality of English LanguageMinor editing of English language is required.
Reviewer 3 Report
Comments and Suggestions for Authors
The study introduces a novel method of estimating shoulder joint rotation angles using machine learning and MediaPipe, which allows for estimation from a forward-facing position, overcoming previous limitations.
Although the paper is interesting, it presents some problems with presentation and novelty.
The authors are invited to review the results and discussion sections; in particular, it is suggested that the two sections, which are currently insufficient, be merged.
In particular, given how the results section is now structured, they are left to the free interpretation of the reader, and even in the discussion session, they are treated superficially. Instead, ample space is given to the shortcomings of the work, limitations in movement, low accuracy of the method, and absence of analysis of shoulder movement.
Round 2
Reviewer 1 Report
Comments and Suggestions for Authors
All my questions have been well addressed. I recommend to accept now.
Comments on the Quality of English LanguageMinor editing of English language required
Reviewer 3 Report
Comments and Suggestions for Authors
The authors followed the reviewer's suggestions.
the main problem of the paper remains the lack of scientific nature of the proposed work, in terms of metrological traceability. A comparison with optoelectronic and/or inertial systems to validate the model would have been very useful.
as minor, the measurement units must be separated from the numerical part
